# Unveiling of the Co-Infection of Peste des Petits Ruminants Virus and Caprine Enterovirus in Goat Herds with Severe Diarrhea in China

**DOI:** 10.3390/v16060986

**Published:** 2024-06-19

**Authors:** Qun Zhang, Xuebo Zheng, Fan Zhang, Xuyuan Cui, Naitian Yan, Junying Hu, Yidi Guo, Xinping Wang

**Affiliations:** State Key Laboratory for Diagnosis and Treatment of Severe Zoonotic Infectious Diseases, Key Laboratory for Zoonosis Research of the Ministry of Education, Institute of Zoonosis, College of Veterinary Medicine, Jilin University, Changchun 130012, China; zhangqun21@mails.jlu.edu.cn (Q.Z.); zhengxb22@mails.jlu.edu.cn (X.Z.); fzhang19@mails.jlu.edu.cn (F.Z.); cuixy22@mails.jlu.edu.cn (X.C.); yannt9920@mails.jlu.edu.cn (N.Y.); jyhu19@mails.jlu.edu.cn (J.H.); guoyd@jlu.edu.cn (Y.G.)

**Keywords:** caprine enterovirus, peste des petits ruminants virus, co-infection, tissue tropism, epidemiological investigation

## Abstract

Here, we report the discovery of two viruses associated with a disease characterized by severe diarrhea on a large-scale goat farm in Jilin province. Electron Microscopy observations revealed two kinds of virus particles with the sizes of 150–210 nm and 20–30 nm, respectively. Detection of 276 fecal specimens from the diseased herds showed the extensive infection of peste des petits ruminants virus (63.77%, 176/276) and caprine enterovirus (76.81%, 212/276), with a co-infection rate of 57.97% (160/276). These results were partially validated with RT-PCR, where all five PPRV-positive and CEV-positive specimens yielded the expected size of fragments, respectively, while no fragments were amplified from PPRV-negative and CEV-negative specimens. Moreover, corresponding PPRV and CEV fragments were amplified in PPRV and CEV double-positive specimens. Histopathological examinations revealed severe microscopic lesions such as degeneration, necrosis, and detachment of epithelial cells in the bronchioles and intestine. An immunohistochemistry assay detected PPRV antigens in bronchioles, cartilage tissue, intestine, and lymph nodes. Simultaneously, caprine enterovirus antigens were detected in lung, kidney, and intestinal tissues from the goats infected by the peste des petits ruminants virus. These results demonstrated the co-infection of peste des petits ruminants virus with caprine enterovirus in goats, revealing the tissue tropism for these two viruses, thus laying a basis for the future diagnosis, prevention, and epidemiological survey for these two virus infections.

## 1. Introduction

Peste des petits ruminants (PPR) is a notifiable disease characterized by pyrexia, diarrhea, pneumonia, ocular and nasal discharge, and oral ulceration [1]. As an acute and highly contagious viral disease, PPR mainly affects small ruminants such as sheep and goats [1]. An outbreak of PPR in goat herds results in a fatality rate of 50–100%, causing huge economic losses to the goat industry [2,3]. PPRV is a single-stranded, negative sense RNA virus that belongs to the genus of *Morbillivirus* within the family of *Paramyxoviridae* [3]. Since first reported in 1942 [4], PPR was primarily prevalent in Western Africa [5,6], the Middle East [7,8], Arabia [9,10], and the South Asian countries [11] in recent years. PPR was first reported in China in July 2007, and is prevalent in more than 20 provinces so far [12,13,14], posing a severe threat to the goat industry.

Enterovirus (EV) is an important pathogen causing a broad array of diseases, including central nervous system infections and digestive and respiratory tract infections in humans and animals [15]. According to the latest virus taxonomy, EV belongs to the genus *Enterovirus* within the family *Picornaviridae* [16]. Currently, the genus of *Enterovirus* contains 12 species of enteroviruses (A-L) and 3 species of rhinoviruses (A, B, C) [16]. Out of 12 enterovirus species, enterovirus E (EV-E), enterovirus F (EV-F), and enterovirus G (EV-G) are mainly related to the infections affecting the livestock industry [17,18,19,20]. While enterovirus infections have been increasingly reported in cattle and pigs, they are still largely unknown in small ruminants such as sheep or goats. In 2012, Boros et al. reported an ovine enterovirus strain OEV-TB4, which was believed to be a strain resulting from the natural interspecies recombination of bovine enterovirus (BEV) strain with porcine enterovirus [21]. In 2017, Wang et al. reported the isolation of a caprine enterovirus (CEV) strain designated as CEV-JL14 from a goat herd manifesting severe diarrhea and respiratory signs with morbidity and mortality rates up to 50–80%, which causes significant economic losses [22].

Mixed infection by different pathogens is commonly observed in clinical practice, which increases the difficulty in precisive diagnosis and prevention of epidemic diseases [23]. PPRV was reported to co-infect goats/sheep with other pathogens such as orf virus [24], border disease virus [25], adenovirus [26], bluetongue virus [27,28], foot and mouth disease virus [29,30], and goat poxvirus [30,31,32,33]. The co-infections usually lead to severe morbidity and mortality in goats [33].

In this study, we report and demonstrate the co-infection of PPRV with CEV that was associated with a disease outbreak characterized by severe diarrhea and respiratory signs on a large-scale goat farm.

## 2. Materials and Methods

### 2.1. Outbreak of a Disease Characterized by Severe Diarrhea and Respiratory Signs

In the middle of October 2018, a disease characterized by severe watery diarrhea, cough, and dyspnea occurred on a large-scale goat farm with approximately 3500 goats in Changchun (Geographic coordinates: 43°88′ N, 125°35′ E), Jilin Province. Epidemiological investigations showed that around 500 goats purchased from Inner Mongolia were introduced to the farm and mixed with the goats on hand. After seven to ten days, some of the original goats manifested clinical signs, such as severe watery diarrhea, cough, dyspnea, and nasal discharges (Figure 1A,B). Within a week, roughly 60% of goats on the farm showed clinical signs at different levels. No PPRV vaccine was used for the goat herds.

Fecal specimens from 276 goats with or without clinical signs were randomly collected and processed for examinations. Fecal specimens were diluted with phosphate-buffered saline (0.01 mol/L, pH 7.2) at 1:5 (*w*/*v*) and were centrifuged at 8000 r/min for 20 min. The supernatants were collected for subsequent experiments.

### 2.2. PPRV and CEV Specimens Controls

PPRV-positive and PPRV-negative controls were collected from PPRV-naturally infected goat herds and healthy goat herds in Gongzhuling City (Geographic coordinates: 43°50′ N, 124°82′ E), Jilin Province, confirmed using RT-PCR and sequencing, and kept in our laboratory [34]. CEV-positive and CEV-negative controls were collected from goat herds with or without CEV infection in Siping City (Geographic coordinates: 43°17′ N, 124°37′ E), Jilin Province [22].

### 2.3. Detection of PPRV and CEV by Electron Microscopy and ELISA

The processed specimens were incubated with 1% phosphotungstic acid for EM examination using an electron microscope (JEM-2200FS/CR) (JEOL, Tokyo, Japan) as previously described [17].

PPRV detection was performed using a commercial ELISA detection kit for PPRV following manufacturer instructions (ID.VET, Montpellier, France). Detection of CEV antigen was parallelly conducted using the CEV antigen detection kit as developed previously [35].

### 2.4. RNA Extraction, cDNA Synthesis and PCR Amplification

Total RNAs extracted from fecal specimens using a TRNzol kit (Tiangen, Beijing, China) were used for cDNA synthesis with SuperScript^TM^ Ⅱ Reverse Transcriptase (Invitrogen, Carlsbad, CA, USA) following the manufacturer’s instruction. PCR was performed using Taq DNA polymerase (Takara, Dalian, China). The PPRV-N gene and CEV-5′UTR were amplified following a program of denaturing the reaction mixture at 95 °C for 5 min, followed by 36 cycles of denaturation at 95 °C for 30 s, annealing at 56 °C for 30 s, and extension at 72 °C for 40 s with a final extension at 72 °C for 10 min.

The primer sequences were listed as follows. PPRV-N-UP: 5′-CACCCGGGCAATTGATACAGC-3′; PPRV-N-DN: 5′-CATGAACCGCCGGAGTGATAGAT-3′. CEV-5′UTR-UP: 5′-GCGGTAGTGCTTTGGTTT-3′; and CEV-5′UTR-DN: 5′-TACGACGTAGCAACACTGGATT-3′.

### 2.5. Histopathological Examinations and Immunohistochemistry Assay

Tissues for histopathological examinations were processed following the standard procedure as previously reported [36]. Tissues fixed in formalin were routinely processed and embedded in paraffin. The embedded blocks were sectioned at 5 μm thickness using a microtome (Leica, Nussloch, Germany). Tissue sections were stained with haematoxylin and eosin (H&E) following the manufacturer’s instructions, examined, and recorded.

An immunohistochemistry assay was performed as previously described [36]. Slides were dewaxed and hydrated before being boiled for antigen retrieval in citrate buffer (pH 6.0) for 15 min. After being treated in 3% H_2_O_2_ for 10 min, the slides were blocked in 5% nonfat milk, and then were incubated with rabbit polyclonal antibodies against PPRV-N protein (Genscript, Nanjing, China) and CEV-encoded VP1 protein [35], respectively. HRP-conjugated goat anti-rabbit IgG (1:1000, Sigma, St. Louis, MO, USA) was used as a secondary antibody. The signal was visualized by the incubation of slides with 3-amino-9-ethyl-carbazole (AEC), and was captured with a CCD camera mounted on a Nikon epifluorescence microscope (Nikon Instruments Co., Ltd., Shanghai, China).

## 3. Results

### 3.1. Two Kinds of Virus Particles Were Revealed in the Specimens Using Electron Microscopy

Fecal specimens from four goats with watery diarrhea were initially examined using EM. Two kinds of virus particles with sizes of 150–210 nm (Figure 1C) and 20–30 nm (Figure 1D) were observed in all four specimens. In terms of the epidemiology, clinical characteristics, and EM examinations, the outbreak of the disease was suspected to be related to PPRV (Figure 1C) and CEV (Figure 1D) infection.

### 3.2. PPRV and CEV Infection Revealed in Goat Herds Using ELISA Methods

To determine viral infection, 276 randomly collected specimens were parallelly analyzed for PPRV and CEV by employing the commercial ELISA kit for PPRV (ID.VET) and an established ELISA kit for CEV [35]. It turned out that 63.77% (176/276) of goats were PPRV-positive and 76.81% (212/276) were CEV-positive, suggesting the intensive infection of PPRV and CEV in goat herds (Figure 1E). Analyses of the detection results showed that 57.97% (160/276) of goats were positive for both PPRV and CEV (Figure 1E), demonstrating the co-infection of PPRV with CEV.

To validate the above results, 25 specimens were analyzed using RT-PCR. All five ELISA-detected PPRV-positive and CEV-positive specimens yielded the expected size of fragments, respectively, while no fragments were amplified from PPRV-negative and CEV-negative specimens (Figure 1F,G). Moreover, the corresponding PPRV and CEV fragments were simultaneously amplified in PPRV and CEV double-positive specimens (Figure 1H), confirming the co-infection of PPRV and CEV.

### 3.3. Histopathological Alterations in Goats Co-Infected with PPRV and CEV

Tissues including lung, kidney, jejunum, and mesenteric lymph nodes from goats naturally infected with PPRV and CEV were histopathologically examined. Obvious lesions were observed in the lung (Figure 2A), including a narrowing of the bronchiole, and the degeneration, necrosis, and desquamation of bronchiolar epithelial cells, with a large number of neutrophilic granulocytes (arrowhead) and exfoliated epithelial cells in the lumen (arrow). In the kidney, fibrinous exudates were seen in the renal capsule. Necrosis and exfoliation of epithelial cells in intestinal glands were also displayed. Hyperplasia of lymphocytes were observed in mesenteric lymph nodes.

### 3.4. Tissue Tropism for PPRV and CEV in Goats Infected with PPRV and CEV

Tissue tropism in goats co-infected with PPRV and CEV was determined using the same set of tissue section ribbons with an immunohistochemistry assay using PPRV-N polyclonal antibody (Genscript) and CEV-VP1 polyclonal antibody [35], respectively. Strong immunoreactions for PPRV were observed in bronchioles, cartilage, jejunum, and mesenteric lymph nodes, while immunoreactions for CEV were observed in the bronchioles, kidney, and intestine (Figure 2B).

Taken together, these results demonstrated the co-infection of PPRV and CEV in goats.

## 4. Discussion

In this study, we discovered the co-infection of PPRV with CEV on a large-scale goat farm characterized by severe watery diarrhea and respiratory signs using different approaches. We found that infection rates of 63.77% (176/276) and 76.81% (212/276) for PPRV and CEV in goats, respectively. These results demonstrated extensive PPRV and CEV infections in the goat herd. More surprisingly, we found that 57.97% (160/276) of goats were co-infected with PPRV and CEV, suggesting that these two viruses were likely to play a synergistic role in causing severe diarrhea and respiratory signs.

PPR and CEV infection are both emerging diseases in China which have been reported to cause a significant economic loss to the goat industry [13,22]. As an emerging disease, PPR was first recorded in Ali of Tibet of China in 2007 [13], and later spread to many provinces due to the trade and distribution of live goats in China in late 2013 and early 2014. Currently, the disease continued to sporadically occur in certain regions. CEV infection was reported to be a novel goat disease characterized by severe diarrhea and respiratory disorder [22], which was ignored or misdiagnosed due to failure in pathogen identification. The isolation of the first caprine enterovirus strain CEV-JL14 makes it possible to perform the investigation on the epidemiology aspects regarding this emerging disease. In early studies, we found that CEV infection was widely detected in goat herds with diarrhea, even in certain goat herds without clinical signs (subclinical infection). The CEV infection rate was significantly higher in diarrheal herds than those of non-diarrheal goat herds [37], suggesting that it is one of the important viruses leading to diarrhea. The findings of high co-infection rate of PPRV with CEV in the goat herds with watery diarrhea confirmed that PPRV was indeed capable of coinfecting goats with CEV. To the best of our knowledge, this is the first report of the mixed infection of PPRV with CEV.

Previous studies showed that PPRV mainly targets the respiratory and digestive system, and PPRV antigens were usually detected in lung, intestine, and lymphoid tissues [38]. The results of our immunohistochemistry assay revealed the strong immunoreactions in the tissues of lung, intestine, and lymph nodes for PPRV antigens, which were congruent with the previous reports [38]. Interestingly, we also found a strong immunoreaction in the cartilage cells for PPRV. To the best of our knowledge, this is the first report of PPRV tropism to cartilage cells. Examination on CEV tissue tropism showed that CEV antigens, like PPRV, were also detected in the lung and intestinal tissues. These results further reaffirmed the co-infection of CEV and PPRV, and supported the likelihood of a synergistic role for PPRV and CEV in the severe diarrhea and respiratory signs. Additionally, strong immunoreactions of CEV in the epithelia cells of the renal distal convoluted tubuli were observed, suggesting that the kidney is likely to be one of the main tissues for CEV tropism. The tropism of CEV on kidney is unclear, which is subject for future investigation.

Tracing or finding the origin of the pathogens is one of the important aspects of the epidemic prevention and controls. Based on our epidemiological investigation, two scenarios regarding the origin of pathogens were proposed. For PPRV, one scenario is that the introduced small herds might harbor some goats of PPRV subclinical infection as reservoirs. After being introduced and mixed with other goats on the farm, these PPRV subclinical infection goats caused the large-scale outbreak. The other scenario is that if these newly introduced herds were healthy goats and served as susceptible herds, the outbreak should be normally limited only to the new susceptible animals. Therefore, we speculated that PPRV probably originated from Inner Mongolia. As for the origin of CEV, we have no solid evidence to confirm the resource, although our previous study found that two sub-genotypes of CEV existed on the same goat farm [39]. Future investigation will focus on the evolution and virulence differences of these CEV strains.

In summary, we have discovered and demonstrated the co-infection of PPRV and CEV by different approaches, and revealed the tissue tropism for these two viruses, which will enrich the understanding of PPRV and CEV infection.

## Figures and Tables

**Figure 1 viruses-16-00986-f001:**
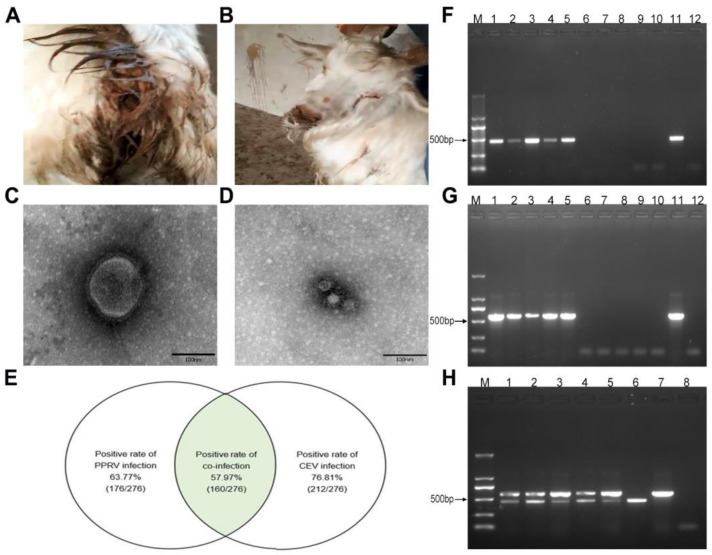
PPRV and CEV co-infection in goats. (**A**,**B**) Representative figures showing the clinical signs for goats characterized by severe watery diarrhea and nasal discharges. (**C**,**D**) Two kinds of virus particles in fecal specimens were observed using electron microscopy with an approximate size of 150 nm (**C**) typical for PPRV and 20–30 nm and (**D**) typical for CEV. Scale bars were marked with a size of 100 nm. (**E**) Venn diagram showing the infection rates of PPRV (63.77%, 176/276), CEV (76.81%, 212/276), and co-infection rate (57.97%, 160/276) of the two viruses revealed using ELISA in a sampling size of 276 goats from a total population of 3500 goats. Co-infection rate is indicated as the green portion. (**F**–**H**) Confirmation of 25 ELISA-detected specimens using RT-PCR. Fragments (452 bp) amplified from PPRV-positive specimens (lane 1–5) with PPRV-negative specimens (lane 6–10) as controls (**F**). M stands for the DNA ladder. Lanes 11–12 refer to positive and negative controls, separately. CEV fragments amplified (**G**). Lanes 1–5, fragments (552 bp) amplified from CEV-positive specimens. Lanes 6–10, results from CEV-negative specimens. PPRV and CEV fragments amplified in ELISA-detected PPRV and CEV double-positive specimens (**H**). Lanes 1–5, fragments amplified from PPRV and CEV double-positive specimens. Lane 6, PPRV-positive control. Lane 7, CEV-positive control. Lane 8, negative control.

**Figure 2 viruses-16-00986-f002:**
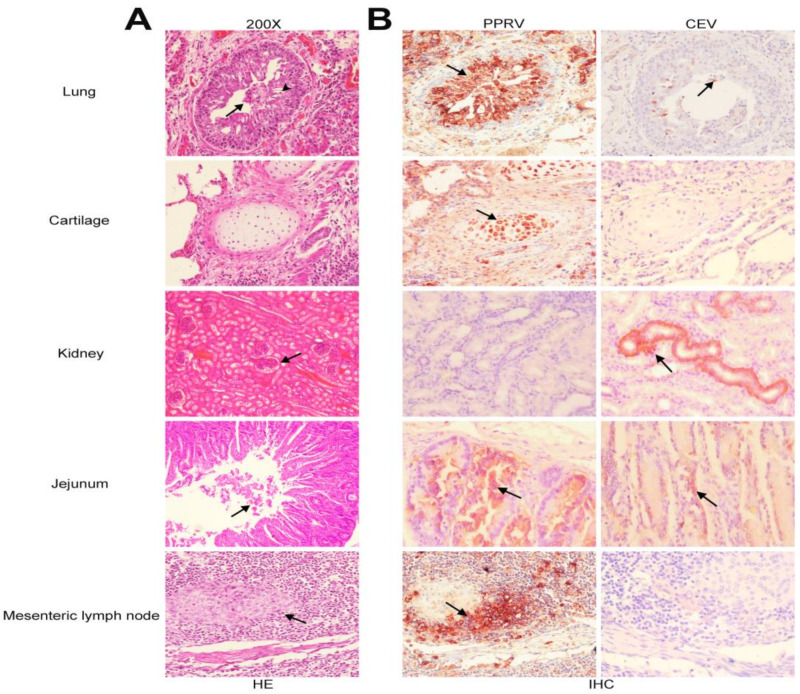
Histopathological lesions and tissue tropism for PPRV and CEV in diseased goats. (**A**) Histopathological lesions revealed hyperplasia of bronchiolar epithelium, scattered epithelial degeneration (arrow), neutrophilic granulocytes (arrowhead) within the lumen, and alveolar septum thickening observed in lung and cartilage cells (the first and second panel from the upper). Fibrinous exudates in the Bowman capsule (arrow) of the kidney (the middle panel). Necrotic and exfoliated epithelial cells (arrow) in the intestinal lumen (the second panel from the bottom). Hyperplasia of lymphocytes (arrow) in the mesenteric lymph node (the last panel). Magnifications were shown in 200×. (**B**) Tissue tropism for PPRV and CEV revealed using the immunohistochemistry assay. PPRV antigens detected in lung, cartilage cells, jejunum, and mesenteric lymph nodes (arrows) using PPRV-N polyclonal antibody; CEV antigens detected in lung, kidney, and jejunum (arrows) using CEV-VP1 polyclonal antibody.

## Data Availability

All data created in this study are available.

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
