# Peer review of "Unveiling of the Co-Infection of Peste des Petits Ruminants Virus and Caprine Enterovirus in Goat Herds with Severe Diarrhea in China"

_viruses, 2024, doi:10.3390/v16060986_

Round 1
Reviewer 1 Report
Comments and Suggestions for Authors
This paper reports the co-infection of PPRV and caprine enterovirus in goats associated with severe diarrhea on a large-scale goat farm in Jilin province,contributing to the enrichment of epidemiological database.However, there are some issues with the content. Here are the specific comments:
1、On line 14, did all of the 276 samples test for at least one virus, and were there any samples that tested negative for both viruses?
2、Did the author detect other viruses associated with diarrhea,such as BVDV?
3、Line 15,130,Figure 1,and so on,all positive rate data should be labelled with the number of positives,for example the infection rates of PPRV (63,76%,,176/276).
4、Previous studies have found that enteroviruses are present in normal populations. So,did the author detect both virus in healthy herds?
5、Paet 3.3 and figer 2,Which section of the small intestine and which lymph nodes should be specified?
Reviewer 2 Report
Comments and Suggestions for Authors
please modify accordingly: see att.
